# Aquatic Invertebrate Community Resilience and Recovery in Response to a Supra-Seasonal Drought in an Ecologically Important Naturally Saline Lake

Lizaan de Necker [1,2,*], Luc Brendonck [1,3], Johan van Vuren [1], Victor Wepener [1] and Nico J. Smit [1]

1   Water Research Group, Unit for Environmental Sciences and Management, North-West University, Private Bag X6001, Potchefstroom 2520, South Africa; Luc.brendonck@kuleuven.be (L.B.); jjvvuren@gmail.com (J.v.V.); victor.wepener@nwu.ac.za (V.W.); nico.smit@nwu.ac.za (N.J.S.)
2   South African Institute for Aquatic Biodiversity (NRF-SAIAB), Makhanda 6139, South Africa
3   Animal Ecology, Global Change and Sustainable Development, Department of Biology, University of Leuven, 3000 Leuven, Belgium
*   Correspondence: Lizaan.denecker@gmail.com or L.denecker@saiab.ac.za

**Abstract:** Climate induced drought is a prominent threat to natural saline aquatic ecosystems by modifying their hydrology and salinity, which impacts the biodiversity of these ecosystems. Lake Nyamithi is a naturally saline lake in South Africa that experienced the effects of a two-year supra-seasonal drought (2015–2016). This study aimed to determine potential effects of the drought and accompanying increased salinity (between 9.8 and 11.5 g $L^{-1}$) on aquatic invertebrate communities of Lake Nyamithi, and assess their potential recovery following the drought. Aquatic invertebrates and water were collected for biodiversity and chemical assessments during predrought conditions (2014), the peak of the drought (2016) and after the site had received water (2017). Taxon richness was considerably reduced during the peak of the drought as many biota could not tolerate the increased salinity. Ecological resilience and recovery was evident in the lake since numerous biota (re)colonized the lake promptly after the site received water and salinity decreased (<8 g $L^{-1}$). By the end of 2017, invertebrate biodiversity exceeded that of predrought conditions. Although some biota may be able to temporarily cope with extreme weather conditions, frequent or prolonged periods of drought and increased salinity pose a threat to naturally saline lakes such as Nyamithi and dilution with fresh water is vital for the persistence of species diversity and ecological integrity.

**Keywords:** climate change; drought; ecological resilience; endorheic lake; salinity; Southern Africa

## 1. Introduction

Naturally saline lakes are found across the globe, in all climatic zones and continents, including Antarctica, although most are confined to arid and semi-arid regions where evaporation exceeds precipitation [1]. These ecosystems represent approximately 44% of the total volume of all lakes on Earth, and include a variety of habitats from permanent salt lakes, including the Caspian Sea, to seasonal and episodically filled ecosystems such as the salt lakes and pans found in Australia and Southern Africa [1–3]. Saline lakes differ from marine waters based on the lack of any physical connection with marine environments, and a more pronounced fluctuation of salinity compared to the relatively stable salinity of marine ecosystems [1,4]. The level of permanence of inland saline ecosystems also affects the long-term and seasonal patterns of salinity, with lower variability in permanent lakes compared to episodic and small, seasonally filled lakes [1,5]. The natural fluctuation in salinity in these ecosystems affects osmotic pressure, ionic composition and the toxicity of ions (to a certain extent), which, in turn, influences the diversity and abundance of biota residing therein [1,4]. Saline lakes are also able to accumulate and recycle nutrients far better than freshwater systems, thus producing large quantities of food for fishes and

birds [3]. These factors create a unique environment with an aquatic and semiaquatic community composition distinct from all other habitats [1,6].

Saline lake ecosystems therefore often have considerable environmental, conservation, social and economic value [1,3,7,8]. Saline lakes attract a wide range of migratory shorebirds and waterfowl for nesting and highly nutritious food in the form of invertebrates and fishes [9–11]. For this reason, many saline lakes are prominent birdwatching sites and a number of them have been designated as Ramsar Wetlands of International importance, including Lake Nyamithi in the northern part of South Africa's KwaZulu-Natal Province [3,12]. However, these environmental and conservation values are often disregarded, as it is difficult to express ecological importance in monetary value [3,9,13,14]. This has led to a global loss of saline lakes due to climate change and anthropogenic impacts [1,3,10,15,16].

The primary threat to the degradation of saline lakes is a change (both reduction or increase) in water inflow as any small alterations in their hydrological budget leads to modifications in salinity and biodiversity [1,3,9,17–19]. Salinity plays a key role in the structuring of aquatic invertebrate communities, particularly once salinity increases >5 g L$^{-1}$ [18,20,21]. Fluctuations either too far above or below an optimal salinity range have been known to result in a loss of biodiversity [20,21]. For this reason, saline lakes are more than other ecosystems susceptible to climate change and anthropogenic alterations to water inflow [1]. It has often been assumed that water flowing into saline lakes is a wasted source and thus many of these ecosystems have been severely affected by a reduction of water inflow due to diversification and abstraction of water for irrigation and domestic use [1,3,9]. The most prominent potential consequences of reduced water inflow in saline lakes are salinization and complete desiccation of permanent saline lakes [9]. Salinization occurs due to secondary salinization through anthropogenic processes (e.g., water abstraction) or as a direct results of drought due to global climate change (e.g., supra-seasonal droughts) [1,21–23], which may eventually also result in total desiccation. Extreme droughts in arid and semi-arid regions such as Southern Africa and Australia are often triggered by strong El Niño Southern Oscillation (ENSO) events as a result of reduced precipitation [1,24]. Although ENSO events are natural phenomena, global climate change may cause them to occur with increasing frequency and intensity. This is of particular concern to semi-arid countries where most saline lakes are found and where the frequency, duration and severity of droughts are predicted to increase greatly [1,9,25].

The resistance and resilience of an aquatic ecosystem towards disturbances such as drought depends on a number of factors including the frequency with which disturbances occur, the length of the disturbance, size and volume of the ecosystem, overall biodiversity and the ability of the aquatic communities to return to predisturbance conditions following the disturbance [26–29]. Larger ecosystems that often experience disturbances are often more resilient to disturbances than ecosystem that do not experience such events [29,30]. Resilience and resistance of aquatic biota is driven by their ability to tolerate, acclimatize or avoid environmental changes [23,31,32]. The length of time necessary for recovery to take place is a measure of the ecosystems resilience [27,33] and occurs rapidly after a disturbance through hatching from resting egg banks in zooplankton, and recolonization by aquatic insects through active dispersal from refuges [27,34]. These mechanisms are an essential part of recovery by aquatic invertebrates after a disturbance [35,36]. The level of recovery depends on duration, intensity and frequency of the disturbance [37–39] as aquatic biota have shown greater resilience to more predictable seasonal or annual droughts while having a variable response to unpredictable supra-seasonal drought [28,32,40].

Aquatic and semiaquatic biota residing in saline lakes are highly adapted to osmotic stress and, if the habitat becomes unfavorable through drying or increased salinity, are able to disperse to more favorable refuges or employ other mechanisms such as the production of resistant dormant life stages to bridge the unfavorable periods [1,35,41,42]. In general, aquatic biota of moderately saline lakes (salinity $\leq$ 10 g L$^{-1}$) consists predominantly of halotolerant freshwater species, including many invertebrates with the ability to

actively disperse or produce desiccation resistant life stages [6,43,44]. As salinity increases, species richness usually decreases with replacement of halotolerant species by halophilic species [1,6,43,44]. However, as there is a salinity threshold above which few, if any, biota are able to survive [20,45], hypersalinization is a serious threat to the biodiversity of moderately saline lakes. The assessment of species richness through a disturbance event is a useful indicator of community resilience [46,47]. Such information may assist researchers in comprehending the potential impacts of disturbance events on an aquatic ecosystem, and support decision and policymakers in the formulation of appropriate and sustainable management tools.

Although there are numerous inland lakes in Southern Africa, research on their ecology, particularly with regard to their aquatic fauna, is scarce [48]. Lake Nyamithi is a moderately saline lake (mean salinity $\pm 3$ g L$^{-1}$) located in Ndumo Game Reserve (NGR) on the lower Phongolo River floodplain (PRF), in Northern KwaZulu-Natal (Northern KZN; South Africa) [12,49]. The lake receives water predominantly from the Phongolo River through an annual controlled flood release in October from the Pongolapoort Dam located approximately 80 km upstream of the game reserve [50,51]. From 2015 to 2016, South Africa experienced the combined effects of a strong ENSO event and a severe supra-seasonal drought, leading to below average rainfall and decreased dam levels across the country, including Northern KZN [12,51–53]. As a result, no floods were released from the Pongolapoort Dam and Lake Nyamithi started to dry up in 2016, with a drop of approximately 1.5 m in water levels, and salinity increasing to levels above 6 g L$^{-1}$ (pers. observation. L. de Necker). This provided an excellent opportunity to assess the effects of an ENSO event on the aquatic communities of one of the largest naturally saline lakes in South Africa and improve the knowledge on its aquatic fauna. Although the ENSO event and drought officially ended in 2017, many regions of South Africa, including Northern KZN, still experience below average rainfall and < 50% dam levels (obtained from: https://www.dws.gov.za/Hydrology/Provincial%20Rain/Default.aspx (accessed on 15 January 2021)) and at the time of writing, the PRF had received no flood releases from the Pongolapoort Dam.

The main aim of this study was to evaluate the effects of a supra-seasonal drought and accompanying changes in salinity on the aquatic invertebrate community of Lake Nyamithi, a naturally saline lake, with special attention for the potential of recovery of the aquatic invertebrates following the drought. This was achieved by comparing aquatic invertebrate communities between predrought, drought and post-drought conditions. It was hypothesized that aquatic invertebrate species richness would significantly decline in Lake Nyamithi during the supra-seasonal drought compared to predrought conditions as a result of increased salinity. It was further anticipated that the aquatic invertebrate community would exhibit resilience and rapid recovery, through recolonization from an egg bank and/or from regional refuges, following the introduction of fresh water and resultant decrease in salinity.

## 2. Materials and Methods

### 2.1. Study Area

The PRF is located on a marine cretaceous bed, which leads to naturally saline groundwater that seeps into the water table and Lake Nyamithi [12,48,49,54]. The lake is situated in NGR, a small reserve (11,898 ha) established in 1924 with the aim to conserve hippopotami (*Hippopotamus amphibius*) and the floodplain's remarkable biodiversity. This game reserve was declared a Ramsar site in 1997 under the Wetlands of International Importance convention as it features five distinct wetland types (including a saline lake) and meets six of the Wetlands of International Importance Criteria [55]. The reserve is also an internationally recognized important bird area (IBA) as it boasts more than 400 bird species [55,56]. Lake Nyamithi is an ecologically important ecosystem as at least 59 of the 120 wetland dependent bird species found in the region have been recorded around the site [55,57]. The lake also houses many hippopotami and one of the largest populations of Nile Crocodile (*Crocodylus*

*niloticus*) in South Africa with an average of 800 individuals [56,58]. Although the main source of water for Lake Nyamithi is flood water from the Pongolapoort Dam, the site also receives water through local rainfall during the areas wet season (November to March), as it has its own catchment, and water from natural floods from the Usuthu River [12,56]. The natural dry season for this region is generally from May to September as there is reduced rainfall and no flood releases. Lake Nyamithi has an area of 183.4 ha [49], the littoral zone consisting predominantly of sparse marginal vegetation including reeds, sedges and some submerged vegetation in the shallow areas (< 30 cm depth) while the benthic area is a fine to coarse sandy substrate. The limnetic zone is represented largely by open-water regions between 30 cm and up to 4 m in depth with isolated patches of aquatic vegetation and a muddy/clay substrate [56].

Four areas were selected for sampling along the shores of Lake Nyamithi to ensure an integrated sampling effort representative of the two zones, vegetation and substrate of the lake (hereafter referred to collectively as biotopes) (Figure 1). Nine sampling surveys were undertaken from 2014 to 2017 during what is considered as the predrought (2014), drought (2016) and recovery (2017) periods. Two surveys took place under predrought conditions with normal rainfall and flooding regimes. The first of these occurred in the natural dry season (September 2014) and the second following a controlled flood release from the Pongolapoort Dam (December 2014). During the peak of the drought in 2016, the site was sampled three times (May, August and December 2016). Four sampling surveys took place in 2017 (February, May, August and November 2017) in what was considered the "recovery" period as Lake Nyamithi continuously received water through isolated rainfall in its catchment and small natural floods from the Usuthu River throughout 2017. Each area was sampled once as part of each survey and the results for each survey collated.

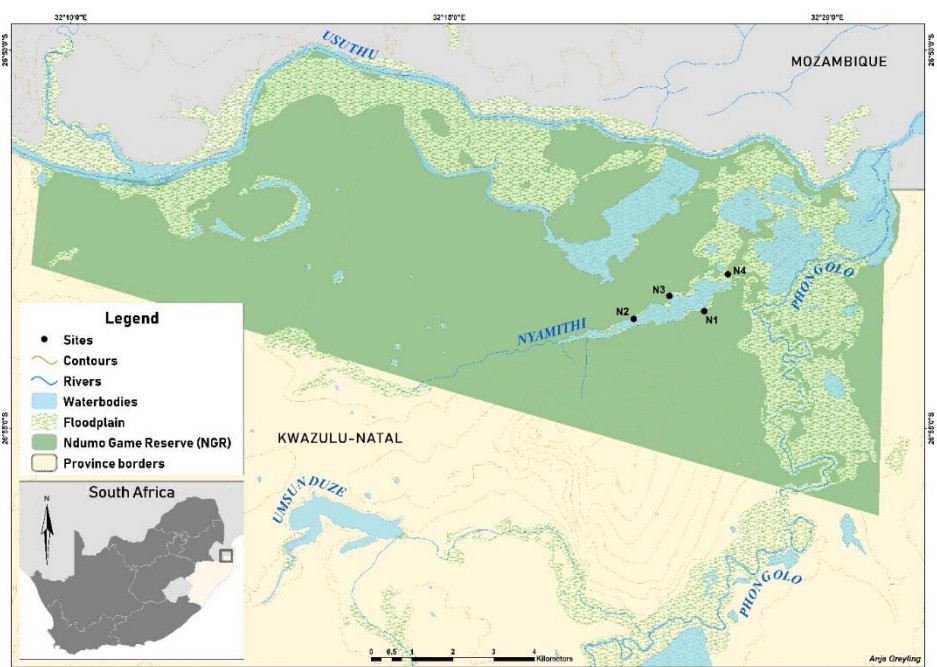

**Figure 1.** Map of Ndumo Game Reserve (NGR) and surrounding region indicating the sampling areas along the shoreline of Lake Nyamithi (N1–N4). N–Nyamithi.

*2.2. Field Sampling*

Physicochemical water variables, namely pH, temperature (°C), dissolved oxygen (mg L$^{-1}$), oxygen percentage saturation (%) and salinity (g L$^{-1}$), were measured in situ at each site with the use of ExStik EC 500 and ExStik DO600 water quality meters (Extech instruments, FLIR Systems, MA, USA). A single, integrated subsurface water sample was then collected from each area in each survey in a sterile bottle and frozen for further analysis.

Aquatic macroinvertebrates were collected across the entire reach of each sampling area to a maximum depth of 50 cm from the available biotopes using a standard 30 cm × 30 cm D-frame dip net. Forty sweeps were taken with the net at each sampling area in each of the nine surveys from the different biotopes available in the area and the integrated sample then placed in a sterile jar and fixed using 5% buffered neutral formalin (NBF). A single integrated zooplankton sample was collected at each sampling area in each of the nine surveys by filtering 80 L of water sampled from different biotopes in a 3–5 m transect through a 64 μm mesh net and stored in 70% ethanol. For safety reasons, due to the presence and abundance of crocodile and hippopotami in Lake Nyamithi, it was not possible to sample invertebrate communities at a depth exceeding 50 cm. Due to the natural saline nature of Lake Nyamithi, little aquatic and marginal vegetation is present in the lake with the majority thereof growing on the edges of the lake while deeper areas contain no floating, submerged or emergent vegetation. Although aquatic invertebrates are found throughout an ecosystem, the vast majority prefer to live in well vegetated habitats as it provides structure for shelter from predation and attachment sites algae and detritus to grow, which is an important food source for many invertebrates [59]. A much higher diversity of aquatic invertebrates is thus found in these areas compared to open-water areas with little to no vegetation [60–62]. Therefore, although it is possible that some invertebrates were not collected due to limitation of not sampling at a depth greater than 50 cm, we are confident that the majority of aquatic invertebrates were collected in our study as the vegetated habitats were thoroughly sampled. All field samples were collected and processed with the necessary permits (Permit No's: OP 899/2016 and OP 1075/2017) and ethical clearance (Ethics No: NWU-00264-16-A5).

*2.3. Laboratory Analysis*

Water chemistry was measured in the laboratory on defrosted, unfiltered water samples using appropriate test kits and standard protocols (test kit catalogue numbers in parentheses) with a Merck Spectroquant Pharo 300 Spectrophotometer (Merck, E Merk KG, Darmstadt, Germany) [63]. Variables assessed included $NO_3^-$ ($NO_3^-$-N) (14773), $NO_2^-$ ($NO_2^-$-N) (14776), $SO_4^{2-}$ (14791), $NH_4^+$ ($NH_4^+$-N) (14752), $NH_3$ ($NH_3$-N) (14752), $Cl^-$ (14897), $PO_4^{3-}$ (orthophosphates) (14848) and total hardness (TH) (100961). Aquatic macroinvertebrate samples were gently rinsed under running tap water to remove the NBF and placed in 70% ethanol. Aquatic macroinvertebrate and zooplankton samples were identified to lowest possible taxonomic level with the use of a Nikon Model SMZ445 C-LEDS dissection microscope (Nikon Corporation, Konan, Minato-ku, Tokyo, Japan) and standard identification guides [64–72] and counted.

*2.4. Data Analysis*

Data analyses on the aquatic invertebrate communities took place on lowest taxon level. Aquatic invertebrate data for each survey were collated into a single result to provide a holistic representation of invertebrate community structure of Lake Nyamithi for each survey. Data were then square root ($\sqrt{}$) transformed prior to analysis to reduce the effect of dominant taxa [73,74]. Analysis of similarity (ANOSIM) was used to test for potential significant differences between aquatic invertebrate assemblages between the nine surveys using the a priori factor "year of sampling survey" ("Year") and visualized using a non-metric multidimensional scaling (nMDS) plot and dendrogram. Similarity percentage analysis (SIMPER) was also used to identify the aquatic invertebrate biota that characterized the community in percentage (%) for each of the three hydrologic periods (predrought, drought and recovery). Diversity indices were calculated from untransformed data and diversity assessed for each survey using Margalef's species richness (richness) [75], Pielou's evenness index (evenness) [76], Shannon–Wiener diversity index (Shannon diversity) [77] and Simpson's index [78]. An ordinary one-way Analysis of Variance (ANOVA) analysis was performed with a Tukey's multiple comparison to determine if there were significant differences in these diversity measures between the hydrologic periods (predrought, drought

and recovery). The ANOSIM, nMDS, SIMPER and diversity indices were completed in the Primer software package with the PERMANOVA+ add-on (Version 7; PRIMER-E Ltd., Devon, UK [79]).

Environmental data (water quality variables) were collated for each survey as with the invertebrate data and, with the exception of pH, log transformed (y=log (x+1)), prior to further analysis to improve data normality. A variance inflation factor (vif) analysis was used to assess possible multicollinearity between water quality variables using the car package, created for the R statistical programme [80,81]. The variables $NO_3^-$, $NO_2^-$ and $NH_3$ showed high multicollinearity with other variables (>10 [82]) and were thus excluded from further analysis. To illustrate the relationships between invertebrate communities (response variables) and environmental data (explanatory variables) between each of the nine surveys, a constrained unimodal canonical correspondence analysis (CCA) was used. A unimodal response model was selected above the linear redundancy analysis (RDA) model as the gradient length of the ordination axes (estimated in turnover or standard deviation) units was > 4 species turnover units (SD) justifying the use of a CCA [83]. Only the 30 best fitting taxa were visualized on the CCA as the multipatt analysis in the indicspecies package for R [81,84] determined that no single aquatic invertebrate taxon was significantly associated with a particular sampling survey. Forward selection was also used as part of the CCA to assess whether any of the remaining water variables significantly explained aquatic invertebrate community structure. Only five chemical water variables (temperature, salinity, total hardness, $PO_4^{3-}$ and $NH_4^+$) were including in the CCA after forward selection. The CCA was created using the Canoco software package (Version 5; Canoco 5, Cambridge, UK [83]).

## 3. Results

From 2014 to 2017 a total of 98 aquatic invertebrate taxa representing 47 families and two superfamilies were collected and identified from Lake Nyamithi (Table S1). The greatest diversity was found in 2017 with 77 taxa, followed by 2014 with 42 taxa while the lowest diversity was reported in 2016 with 23 taxa. Individuals of the families Ceratopogonidae (*Bezzia* sp.), Chironomidae (Chironominae), Corbiculidae (*Corbicula fluminalis*), Culicidae, Dytiscidae and Libellulidae were present in varying numbers in at least one survey in each of the three hydrologic periods (predrought, drought and recovery) while individuals of families Corixidae (*Agraptocorixa* sp., *Micronecta* sp. and *Sigara* sp.) and Hydrophilidae (*Berosus* sp. and *Hydrophilus senegalensis*) were present in all surveys in varying abundance (Tables S1 and S2). Ostracoda (including the superfamilies Cypridoidea and Darwinulidae) were present in all but the August 2016 survey, and was often the most abundant taxon per survey with the most individuals collected in February 2017 (3471 individuals). The most abundant taxa in September and December 2014 were the invasive snail *Tarebia granifera* (2277 and 321 individuals, respectively) and the native mollusks *Bulinus natalensis* (148 and 1134 individuals, respectively) and *B. tropicus* (382 and 266 individuals, respectively). These biota were also present only in the predrought surveys of 2014 while fewer genera of snail in much lower abundance were present during the 2016 and 2017 surveys and included *Cleopatra* sp. (three individuals in May 2017), *Corbicula* sp. (four individuals in August 2016; three individuals in August 2017) and another invasive snail, *Physa acuta* (93 individuals in May 2017). Harpacticoida was the most abundant taxon in 2016 with 810 individuals collected in August 2016 (810 individuals) followed by *Sigara* sp. (464 individuals in August 2016) while *Diaphanosoma* sp. and *Thermocyclops* sp. were in highest abundance in February 2017 (420 and 960 individuals, respectively). These four abovementioned zooplankton were also present only during the three drought (2016) and four recovery (2017) surveys (Tables S1 and S2).

Two groupings were evident based on the hierarchical cluster dendrogram with invertebrate taxa from the three 2016 (drought) surveys separating from the 2014 and 2017 (predrought and recovery respectively) surveys at 24% similarity (Figure 2a). An additional separation occurred between the two 2014 surveys (predrought) and the four

2017 surveys (recovery) at 32% similarity. At a stress level of 0.05 the nMDS indicated a good fit ordination and a similar pattern to the dendrogram was observed as each year separated distinctly from the rest (Figure 2b). The most similar groupings (>40%) were the two predrought surveys, September and December 2014 (50.12%), followed by recovery surveys February and May 2017 (43.63%), and August and November 2017 (42.57%) and finally two of the drought surveys August and December 2016 (42.35%). Using the preselected factor "Year", the ANOSIM indicated significant differences between these groups observed on the dendrogram and nMDS plots (Global R = 0.91; significance level = 0.2%). Results from the SIMPER analysis indicated that the two predrought surveys (2014) were 50.12% similar and the invertebrate taxa responsible for this were Ostracoda, *T. granifera*, *B. tropicus*, *B. natalensis*, *Agraptocorixa* sp. A, Baetidae and *Enallagma* sp. (Table 1). The three drought surveys (2016) were 35.55% similar and the taxa responsible for this similarity were *Sigara* sp. and Harpacticoida. Finally, the four recovery period surveys (2017) were 35.43% similar and the representative taxa responsible were Ostracoda, Chironominae, *Hyphydrus* sp. A, *Appasus* sp., *Thermocyclops* sp., *Pseudagrion* sp. and *Micronecta* sp.

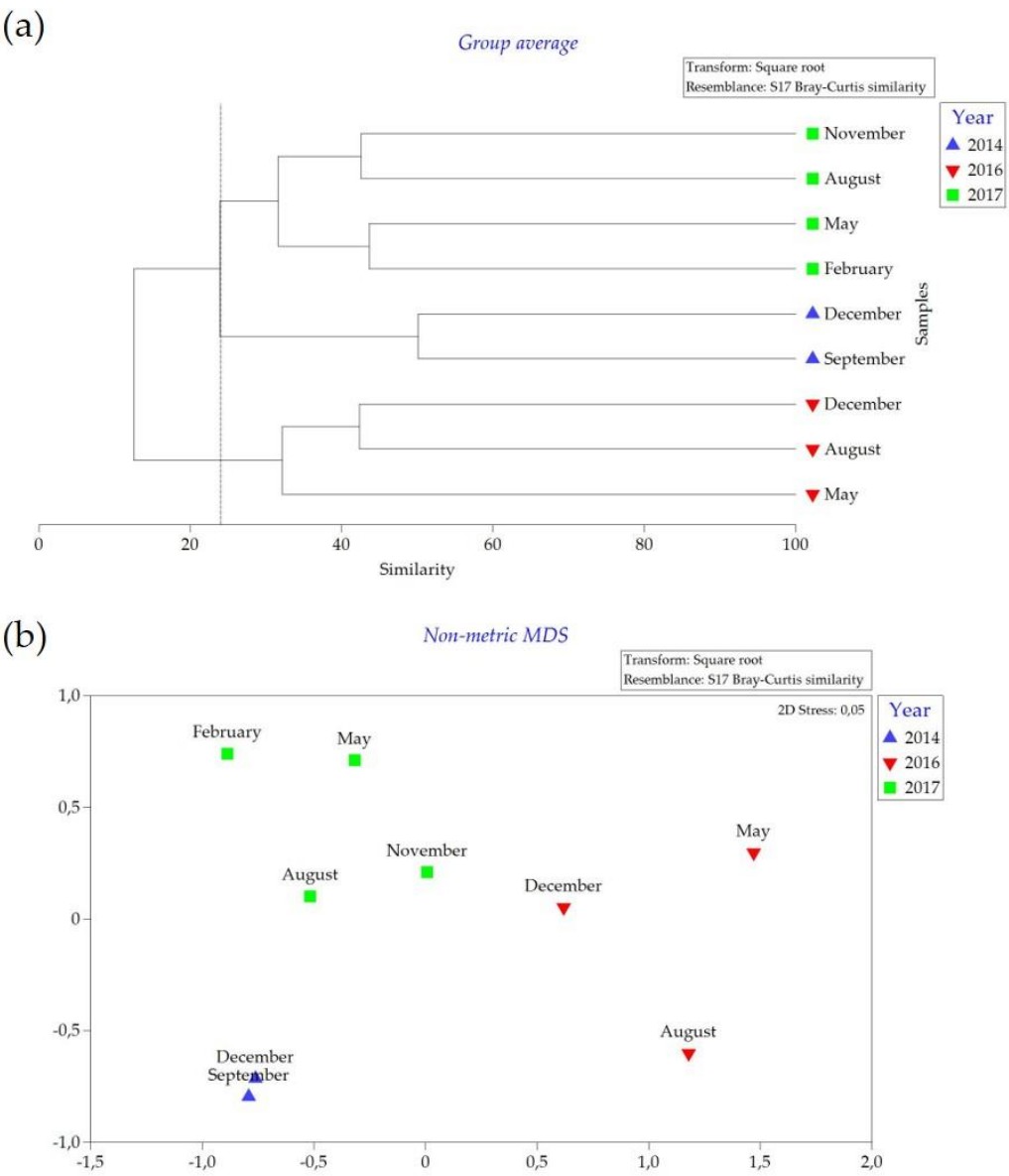

**Figure 2.** (**a**) Dendrogram and (**b**) non-metric multidimensional scaling (nMDS) plots of aquatic invertebrate taxa collected during sampling surveys in 2014 (predrought), 2016 (drought) and 2017 (recovery) in Lake Nyamithi.

**Table 1.** Contribution of various aquatic invertebrate taxa to within group similarities in Lake Nyamithi determined through similarity percentage analysis (SIMPER). Groups are based on the hydrologic period in which each survey took place: 2014 (predrought); 2016 (drought) and 2017 (recovery) and were determined using dendrogram and nMDS analysis.

| Year | Overall Similarity | Species | Average Abundance | Average Similarity | Percentage Contribution (%) | Cumulative Contribution (%) |
|------|------|------|------|------|------|------|
| 2014 | 50.12% | Ostracoda | 22.2 | 7.7 | 15.3 | 15.3 |
|  |  | *Tarebia granifera* | 32.8 | 7.6 | 15.1 | 30.4 |
|  |  | *Bulinus tropicus* | 17.9 | 6.9 | 13.7 | 44.1 |
|  |  | *Bulinus natalensis* | 22.9 | 5.1 | 10.3 | 54.4 |
|  |  | *Agraptocorixa* sp. A | 14.8 | 3.8 | 7.6 | 61.9 |
|  |  | Baetidae | 7.7 | 3 | 6.1 | 68 |
|  |  | *Enallagma* sp. | 7.4 | 2.3 | 4.5 | 72.6 |
| 2016 | 35.55% | *Sigara* sp. | 12.5 | 15.4 | 43.4 | 43.4 |
|  |  | Harpacticoida | 15.5 | 13.3 | 37.3 | 80.8 |
| 2017 | 35.43% | Ostracoda | 26.1 | 9.4 | 26.4 | 26.4 |
|  |  | Chironominae | 5.5 | 3.5 | 10 | 36.4 |
|  |  | *Hyphydrus* sp. A | 7.3 | 3 | 8.4 | 44.8 |
|  |  | *Appasus* sp. | 6.6 | 2.9 | 8.2 | 53 |
|  |  | *Thermocyclops* sp. | 12.4 | 2.7 | 7.7 | 60.7 |
|  |  | *Pseudagrion* sp. | 4 | 2.1 | 5.8 | 66.5 |
|  |  | *Micronecta* sp. | 3 | 1.4 | 4 | 70.5 |

The CCA explained 45.6% of data variation on the first two axes and separation is evident between the three sampling years, particularly between predrought and drought conditions (Figure 3). Variation along axis 1 (explaining 27.8%) was driven predominantly by the large difference in invertebrate diversity and community structure, as evidenced by the differences in the various diversity indices, between predrought and drought conditions including number of taxa, number of individuals and richness, which were much lower in 2016 (drought) compared to 2014 (predrought). The ANOVA analysis indicated these differences were not significant ($p > 0.05$). Furthermore, most of the indices were lowest in May 2016 (drought), namely number of taxa (5), number of individuals (101), richness (0.87), Shannon diversity (0.66) and Simpson diversity (0.37) (Figure 4a–f). These indices were all highest at the end of the recovery period as August 2017 had the greatest number of taxa (59) and richness (7.99), while November 2017 (recovery) had the highest evenness (0.83); Shannon diversity (2.82) and Simpson's index (0.93). Separation along axis 2 (explaining 17.8%) was driven mainly by differences in water quality variables between surveys. May and August 2016 (drought) showed a positive association with total hardness, salinity and $Cl^-$ during the peak of the drought as a result of the highest measured levels of total hardness in May 2016 (1910 mg $L^{-1}$); and salinity (11.5 g $L^{-1}$) and $Cl^-$ (3280 mg $L^{-1}$) in August 2016 (Table 2). These surveys were also negatively associated with temperature as August 2016 (drought) had the coldest measured water temperature (17 °C). Salinity was lowest in December 2014 (predrought) (2.3 g $L^{-1}$), while total hardness and $Cl^-$ had lowest values in February 2017 (recovery) (195 mg $L^{-1}$ and 193 mg $L^{-1}$, respectively) and water temperature was highest in November 2017 (recovery) (33.5 °C, Table 2). September 2014 (predrought) was positively associated with $NH4^+$ with highest concentration during this survey (2.2 mg $L^{-1}$) while August 2017 (recovery) had the highest $PO_4^{3-}$ (4.0 mg $L^{-1}$) along with the remaining nutrients ($NO_3^-$, $NO_2^-$ and $NH_3$) and $SO_4^{2-}$ not shown on the CCA (Table 2).

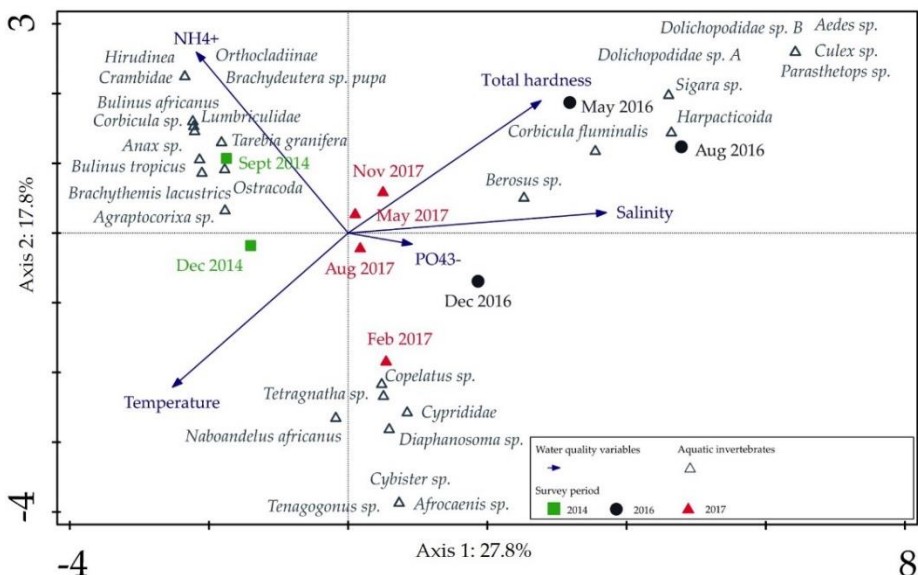

**Figure 3.** Canonical correspondence analysis (CCA) plot comparing aquatic invertebrate diversity and select water quality variables to all sampling surveys during 2014 (predrought), 2016 (drought) and 2017 (recovery) in Lake Nyamithi. The plot explains 45.6% of data variation of which 27.8% is explained by axis 1 and 17.8% by axis 2. Only the 30 best fitting aquatic invertebrate taxa and water quality variables selected through correlation and forward selection analysis are presented.

**Table 2.** Measured water quality variables for each sampling surveys in 2014 (predrought), 2016 (drought) and 2017 (recovery) from Lake Nyamithi.

| Water Quality Variable | September 2014 | December 2014 | May 2016 | August 2016 | December 2016 | February 2017 | May 2017 | August 2017 | November 2017 |
|---|---|---|---|---|---|---|---|---|---|
| Temperature (°C) | 26.1 | 29.6 | 23.4 | 17.6 | 28.5 | 29.7 | 23.8 | 22.9 | 33.5 |
| pH | 8.8 | 7.9 | 8.5 | 9.3 | 8.2 | 8.4 | 7.1 | 8.3 | 8.8 |
| Dissolved Oxygen (mg L$^{-1}$) | 5.8 | 8.3 | 6.0 | 10.3 | 5.5 | 4.5 | 1.9 | 5.3 | * |
| Oxygen Saturation (%) | 72.4 | 105.6 | 72.4 | 105.8 | 70.3 | 56 | 19.5 | 62.5 | * |
| Salinity (g L$^{-1}$) | 3.9 | 2.3 | 9.8 | 11.5 | 10 | 5.4 | 2.3 | 4.9 | 7.1 |
| PO$_4$$^{3-}$ (mg L$^{-1}$) | 0.1 | 0.2 | 0.5 | 0.7 | 0.3 | 0.1 | 0.03 | 4.0 | 0.1 |
| NO$_3$$^-$ (mg L$^{-1}$) | 2.2 | 0.4 | 0.9 | 1.5 | 1.0 | 1.1 | 1.1 | 8.2 | 0.8 |
| NO$_2$$^-$ (mg L$^{-1}$) | 0.04 | 0.04 | 0.02 | 0.01 | 0.04 | <0.02 | 0.02 | 0.3 | * |
| NH$_4$$^+$ (mg L$^{-1}$) | 2.2 | 0.1 | <0.05 | 0.1 | 0.2 | 0.2 | 0.1 | 0.5 | 0.1 |
| NH$_3$ (mg L$^{-1}$) | <0.01 | <0.01 | 0.02 | <0.01 | 0.04 | 0.01 | 0.01 | 0.4 | * |
| SO$_4$$^{2-}$ (mg L$^{-1}$) | 313 | 243 | 975 | * | 536 | 611 | 45 | 2283 | 1320 |
| Cl$^-$ (mg L$^{-1}$) | 731 | 567 | 300 | 3280 | 1068 | 193 | 356 | 1348 | 1720 |
| Total Hardness (mg L$^{-1}$) | 284 | 234 | 1910 | 1130 | 623 | 195 | 188 | 915 | 977 |
| Mg$^{2+}$ (mg L$^{-1}$) | 79 | 51 | 1220 | 660 | 355 | 123 | 124 | 333 | 388 |
| Ca$^{2+}$ (mg L$^{-1}$) | 206 | 183 | 690 | 470 | 268 | 71 | 64 | 581 | 645 |

* Denotes data not available.

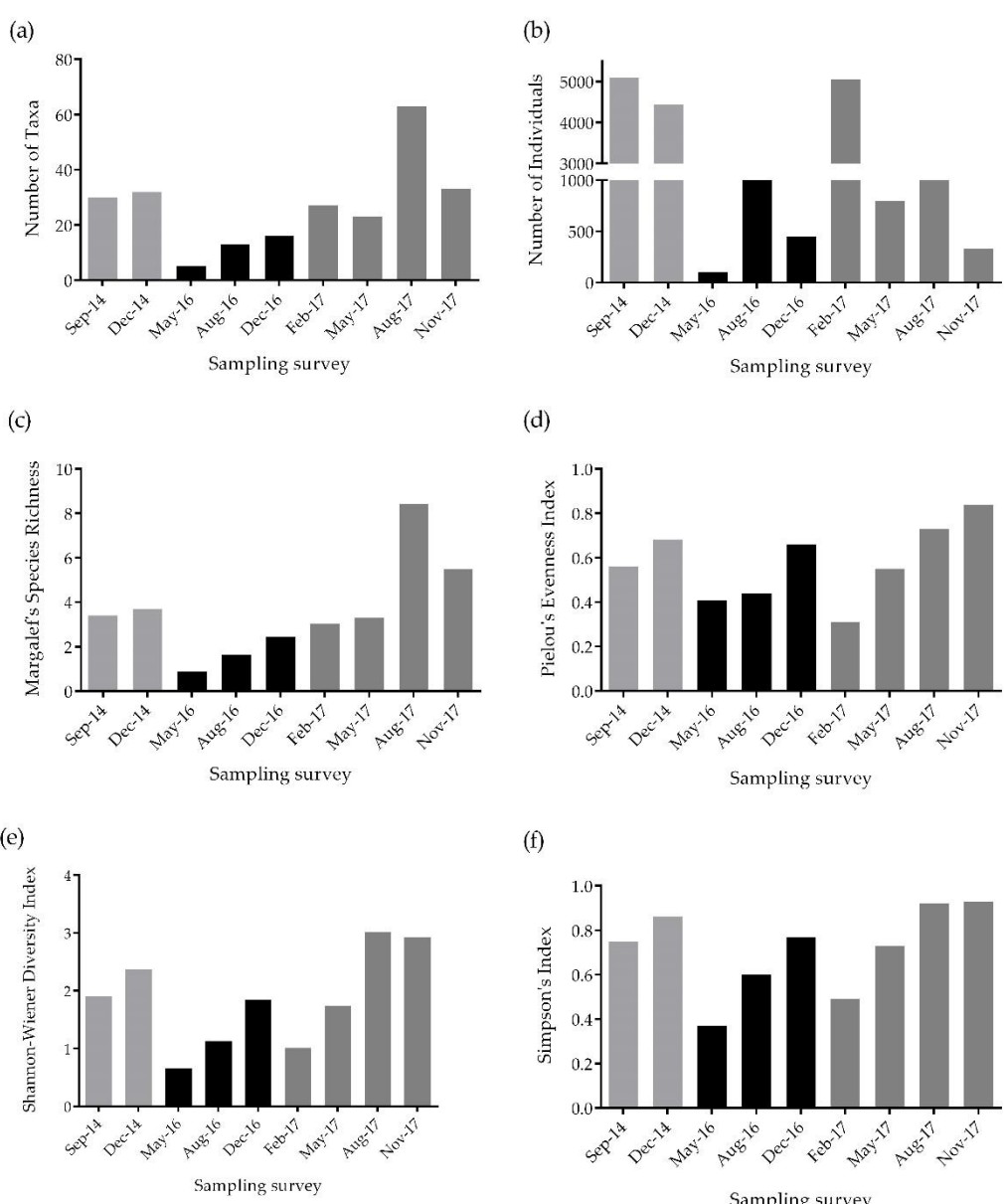

**Figure 4.** Comparison of the number of taxa (**a**), number of individuals (**b**), Margalef's species richness (**c**), Pielou's evenness index (**d**), Shannon–Wiener diversity (**e**) and Simpson's index (**f**) of aquatic invertebrates collected during the sampling surveys in 2014 (predrought), 2016 (drought) and 2017 (recovery) in Lake Nyamithi.

## 4. Discussion

This study evaluated the aquatic invertebrate community of a naturally saline lake in Southern Africa as it experienced the effects of a supra-seasonal drought with increased salinity. The main aim was to assess the effects of the drought by comparing the aquatic invertebrate community to that of predrought and recovery conditions. Abiotic and biotic variables differed greatly between predrought (2014) and drought (2016) conditions with a pronounced reduction in invertebrate diversity and abundance in the drought period when the lake was at its lowest level and most saline. As a result of isolated rainfall and small, localized floods in the recovery period (2017), salinity in the lake decreased and was accompanied by a rapid increase in aquatic invertebrate diversity and abundances, with diversity exceeding that of predrought conditions. These findings provide important information on the effect of a supra-seasonal drought on aquatic invertebrates and demonstrate the importance of sustaining a balance between salinity concentration and fresh water inputs in a saline ecosystem to maintain ecological integrity and high aquatic biodiversity.

As salinity increased $> 9\,\mathrm{g\,L^{-1}}$ during drought conditions (2016) the number of aquatic invertebrate taxa in Lake Nyamithi reduced by approximately half compared to that of predrought conditions (2014). Similar results were obtained by Senner et al. [21] who reported decreased invertebrate numbers in North American saline lakes during periods of decreased water levels and subsequent increased salinity. Among the invertebrates that disappeared from Lake Nyamithi were most mollusks, water scorpions (Nepidae) and burrowing water beetles (Noteridae). The abovementioned taxa reappeared during the recovery (2017), with the exception of the mollusks including the invasive snail *T. granifera*, which remained absent up to the end of the study in November 2017. The extent with which aquatic invertebrate taxonomic reduction occurs during periods of stress, such as drought or increased salinity, depends on the traits and abilities the invertebrates possess to withstand the changing conditions [25,40]. Those with adaptations or life history traits that enable them to tolerate changes in their environmental conditions; produce resting life stages or actively migrate to a nearby ecosystem to avoid the stressor, will survive, while those without these abilities will experience local extinction [40,85]. Water scorpions and burrowing water beetles are aquatic biota with the ability to actively move between ecosystems and likely migrated to nearby ecosystems such as the Phongolo and Usuthu rivers during the drought period (2016). Soft-bodied organisms, including mollusks, are particularly susceptible to desiccation and the accompanying effects, such as increased salinity, and consequently will usually disappear during such periods of stress [42,86,87]. *Tarebia granifera*, which were extremely dominant during predrought conditions, is an immensely invasive molluscan species in Southern Africa. This species may reach densities of several thousand per square meter in invaded areas [88] and has a known tolerance to moderate salinity levels $(10–13\,\mathrm{g\,L^{-1}})$, capable of surviving in hypersaline water $(40\,\mathrm{g\,L^{-1}})$ for up to a month under laboratory conditions [89]. This is most probably the reason that this snail has become a successful invader in South Africa, not only of freshwater systems, but also estuaries, lagoons and coastal lakes [89–91]. However, while *T. granifera* may be able to live in moderate salinity levels in nature, it was likely the combined effects of a prolonged period of increased salinity, along with a reduction in potential food sources that ultimately caused it to be unable to survive through the drought.

Of all invertebrates collected (98 taxa), several families including Baetidae, Ceratopogonidae, Chironomidae, Corixidae, Dytiscidae, Hydrophilidae and Libellulidae, were present in at least one survey in each of the three hydrologic periods (predrought, drought and recovery). Of these, only corixids (water boatmen) and hydrophilids (water scavenger beetles) were present in each survey and ostracods (seed shrimp) in all but one survey. These taxa are predominantly found in freshwater ecosystems, but known to house drought tolerant and halotolerant species that are able to survive droughts and live in highly saline conditions [65,71,92]. Indeed, genera of the families Chironomidae, Corixidae (*Sigara* sp. and *Micronecta* sp.), Hydrophilidae (*Berosus* sp.), Baetidae (*Cloeon* sp. and *Procloeon* sp.) and Dytiscidae are hardy, freshwater macroinvertebrates, exceptionally tolerant to changes in the environment, including pool drying and/or high salinity, and known to occur in various ecosystems from temporary wetlands to saline lakes and streams [17,39,86,92–94]. Similarly, Waterkeyn et al. [17] observed that copepods and corixids (*Sigara* sp. and *Micronecta* sp.) survived salinity up to $25\,\mathrm{g\,L^{-1}}$ in temporary wetlands in France, and Velasco et al. [94] observed species of the genus *Berosus* in high abundance in a Mediterranean saline stream. Various species of Ceratopogonidae (biting midges), Chironomidae (non-biting midges) and Libellulidae (skimmers) also have drought resistant mechanisms to survive temporary periods of drought including aestivating in wet mud (Ceratopogonidae and Libellulidae) and short aquatic larval life stages (Chironomidae) [69,95–97]. Ostracods are also present in most aquatic ecosystems, with genera found in both marine and non-marine environments, and able to tolerate harsh environmental conditions including salinities of up to three times that of seawater [65,98,99]. Harpacticoida (Copepoda) were one of the only invertebrate taxa, apart from corixids and hydrophilids, still present in Lake Nyamithi at the extremely elevated salinity $(>10\,\mathrm{g\,L^{-1}})$ in May 2016 and likely hatched from resting

stages in the sediment in response to the increased salinity. The vast majority of genera of Harpacticoida are estuarine and marine ($\pm 85\%$) and therefore thrive in highly saline environments [100,101]. Once salinity increases to approximately $8 \text{ g L}^{-1}$, there is a decline in halotolerant invertebrates and a concomitant increase in biota found in highly saline to estuarine ecosystems [1,4,42,86].

Salinity in Lake Nyamithi decreased ($<6 \text{ g L}^{-1}$) from December 2016 onwards, following a large rainstorm in its local catchment (pers. observation; L de Necker). Through continued rains and localized floods from the Usuthu River throughout the year, salinity remained lower in the lake throughout the recovery period ($<8 \text{ g L}^{-1}$) than during the peak of the drought (August 2016; $>10 \text{ g L}^{-1}$). A rapid recovery of invertebrates was evident in response to the decrease in salinity during the recovery as several aquatic invertebrates, including invertebrates previously recorded during predrought conditions namely *Anisops* sp. (Notonectidae), *Appasus* sp. (Belostomatidae) and *Micronecta* sp. (Corixidae), *Ranatra* sp. (Nepidae) and *Canthydrus* sp. (Noteridae) swiftly (re)colonized the lake. As these groups are active migrants that can seek refuge under extreme conditions, they likely recolonized the lake from nearby freshwater sources such as the Phongolo and Usuthu Rivers. The rate at which invertebrates reappear after a period of stress is dependent on the traits of the aquatic invertebrates and amount of connectivity between aquatic habitats [102,103]. Biota with resting life stages are stimulated by change in their abiotic environment to re-emerge or hatch first while the more opportunistic organisms that actively disperse will migrate from nearby habitats soon after [103,104].

Previously unrecorded zooplankton such as *Thermocyclops* sp. (Cyclopidae, Copepoda), *Moina micrura* (Moinidae, Cladocera) and *Diaphanosoma* sp. (Sididae, Cladocera) that likely hatched from resting stages in the lake sediment, were also sampled for the first time. By the end of the studied recovery period (November 2017) biodiversity had even surpassed that of the predrought period (2014), even though salinity was still higher in the recovery ($7 \text{ g L}^{-1}$) compared to the predrought ($<4 \text{ g L}^{-1}$) period. This may be related to a similar observation made in other, more saline, environments whereby biodiversity is highest at a level of intermediate salinity (for that particular ecosystem) and deviation from this "optimal" salinity (either above or below) results in a loss of biodiversity [20,21,105]. The increased biodiversity in the recovery (2017) may also be related to the reorganization phase that ecosystems undergo after a destabilizing event such as drought or flood. In this phase, an ecosystem experiences high biodiversity as a result of the influx of new species [106], as was also observed in our study with the presence of several previously unrecorded aquatic invertebrate taxa. Ecosystem productivity also often returns to, or surpasses the "typical" levels in the year following a climate event such as a drought [27]. Biodiversity increases in response to the inflow of fresh water as this stimulates the hatching of zooplankton from resting stages in the sediment, if present, and (re)colonization of active dispersers from nearby habitats. This is particularly true in highly connected ecosystems as it allows for more rapid rates of ecosystem recovery than isolated ecosystems as there are more habitats available to be used as refugia [29,107]. Similarly, in a study of various lakes in Uzbekistan, Ginatullina et al. [23] discovered that zooplankton species richness had increased and surpassed that of predisturbance conditions within a year of being affected by increased salinity. Similarly, in a study of various lakes in Uzbekistan, Ginatullina et al. [23] discovered that zooplankton species richness had increased and surpassed that of predisturbance conditions within a year of being affected by increased salinity.

It is likely this natural wet/dry regime experienced in Lake Nyamithi contributed to the observed rapid recolonization and recovery in the lake. Ecosystems that experience frequent bouts of disturbances such as droughts and floods are more resilient and inclined to rapid recovery [25,29,30,108]. Under normal conditions, Lake Nyamithi and adjacent waterbodies experience seasonal periods of floods and drought as part of the controlled flood regime from Pongolapoort Dam that takes place between October and December, and naturally reduced rainfall between May and September [51]. Thus, many of the invertebrate communities found in these ecosystems consist of taxa with adaptations to overcome

these periods of disturbance and provide more resilience to changes in environmental conditions. These mechanisms of survival and recovery are essential to sustain ecosystem functions and services as aquatic invertebrates, especially zooplankton, are an important food source to other aquatic invertebrates and for fish and birds, particularly in saline lakes [3,10,21,109,110].

Several invertebrates recorded in Lake Nyamithi in the predrought (2014) were still absent by the end of the studied recovery period (2017). These included most mollusks, indicating their inability to recover rapidly from an extended drought period. Some of these snails, including *T. granifera*, *B. natalensis* and *B. tropicus* possess the ability to persist temporarily through periods of drought or increased salinity by burrowing into sand, closing their operculum (if present) or limiting their physiological activity [68,90]. However, the length of the present drought and severity of salinity increase likely created an environment too difficult for many of these snails to survive through. Snails are also limited in the dispersal mechanisms as they are unable to actively disperse and thus rely on other strategies of transport, including eggs that attach to birds and other invertebrates through mud [111]. An unexpected positive outcome resulting from the drought was the continued absence of *T. granifera* from the lake during the recovery period. This species is exceptionally invasive with reported negative effects on other aquatic biota including native snails [12,89,90,112,113]. The local extinction of *T. granifera* may thus facilitate native mollusk species to recolonize or establish in Lake Nyamithi without severe competition. However, it is expected that *T. granifera* will re-establish in Lake Nyamithi along with other native snails once the Lower Phongolo River floods again, since they are still found throughout this system and have therefore not disappeared entirely.

## 5. Conclusions

The supra-seasonal drought and consequent decrease in the water level of Lake Nyamithi led to increased salinity that resulted in a substantial decline in invertebrate species richness, supporting our hypothesis of the negative impact of increased salinity on invertebrate richness. The aquatic invertebrate community structure was altered and species numbers reduced as many invertebrates were seemingly unable to resist and tolerate the much increased salinity (>9 g L$^{-1}$) and thus disappeared as the drought progressed leaving only the most tolerant and halophilic invertebrate taxa. Resilience of Lake Nyamithi was illustrated by the rapid recovery of aquatic invertebrate diversity and abundance following the severe drying of the lake. Biota with aerial dispersal mechanisms and desiccation resistant life stages present in the sediment (re)colonized rapidly in response to the fresh water inputs into the lake. Other taxa, including the invasive snail *T. granifera*, were unable to re-establish during the recovery period as a result of their limited dispersal mechanisms. This alien snail species is, however, not yet eradicated from the ecosystem as they are still present in the Lower Phongolo River.

Through years of constant wet/dry cycles, many of the aquatic biota of Lake Nyamithi have likely adapted to living in a constantly changing environment and are therefore able to survive through, and rapidly recover from, a disturbance such as a relatively short supra-seasonal drought with accompanying salinity changes. However, even though Lake Nyamithi did not experience hypersaline conditions, frequent or prolonged high salinity conditions are still a potential risk to the integrity of the ecosystem. If more frequent droughts, with resulting hypersaline conditions, were to occur for extended periods of time, as predicted due to climate change, it is quite possible that the most tolerant taxa would disappear or that the buffering effect of a mixed egg bank of some resilient taxa would be decreased or exhausted. It is further evident from the results that freshwater inputs form an integral part in sustaining ecological integrity of Lake Nyamithi. It is thus imperative that sustainable management strategies must include actions that will prevent change or reduction in the water supply to this unique ecosystem.

**Supplementary Materials:** The following are available online at https://www.mdpi.com/article/10.3390/w13070948/s1, Table S1: Detailed list of the diversity and abundance of aquatic invertebrates collected in Lake Nyamithi in each sampling survey during the predrought (2014), drought (2016) and recovery (2017) hydrologic periods, Table S2: List of the aquatic invertebrate taxa present in Lake Nyamithi in the predrought (2014), drought (2016) and recovery (2017) hydrologic periods. Various symbols indicate the presence of known halotolerant and drought tolerant invertebrate taxa within the named Family or Order. * contains known halotolerant taxa, ¥ contains known drought tolerant taxa.

**Author Contributions:** Conceptualization: L.d.N., V.W. and N.J.S.; Data curation: L.d.N.; Formal analysis: L.d.N.; Funding acquisition: L.B., V.W. and N.J.S.; Investigation: L.d.N.; Supervision: J.v.V., L.B., V.W. and N.J.S.; Writing—Original draft: L.d.N.; Writing—Review and editing: J.v.V., L.B., V.W. and N.J.S. All authors have read and agreed to the published version of the manuscript.

**Funding:** The financial assistance of the Deutscher Akademischer Austauschdienst (DAAD) and the National Research Foundation (NRF) towards this research is hereby acknowledged (Grant UID 105122). Opinions expressed, and conclusions arrived at, are those of the author and are not necessarily to be attributed to the NRF. L.d.N. acknowledges funding from National Research Foundation (NRF)-Department of Science and Innovation (DSI) Professional Development Programme (Grant UID 127549) and the use of infrastructure and equipment provided by the NRF-SAIAB Research Platform and the funding channeled through the NRF-SAIAB Institutional Support system. This paper forms part of a Water Research Commission of South Africa funded study (Project K5-2185, PI: NJ Smit) and was co-funded by a VLIR-UOS TEAM project (ZEIN21013PR396, PI: L Brendonck and V Wepener).

**Institutional Review Board Statement:** The study was conducted according to the guidelines of the Declaration of Helsinki, and approved by the AnimCare Animal Research Ethics Committee (AREC-130913-015) of North-West University (NWU-00264-16-A5; approved 20 September 2016).

**Informed Consent Statement:** Not Applicable.

**Data Availability Statement:** Data is contained within the article and supplementary material.

**Acknowledgments:** The Department of Agriculture, Forestry and Fisheries (DAFF) of South Africa is hereby acknowledged for providing the data necessary to generate the site and irrigation map for this manuscript. Further, Anja Erasmus is acknowledged for creating the map.

**Conflicts of Interest:** The authors declare no conflict of interest.

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
