# Peer review of "Aquatic Invertebrate Community Resilience and Recovery in Response to a Supra-Seasonal Drought in an Ecologically Important Naturally Saline Lake"

_water, doi:10.3390/w13070948_

Round 1
Reviewer 1 Report
Manuscript “Aquatic invertebrate community resilience and recovery in response to a supra-seasonal drought in an ecologically important naturally saline lake”
by Lizaan de Necker, Johan van Vuren, Luc Brendonck, Victor Wepener and Nico Smit
GENERAL COMMENTS
The manuscript aimed to determine the potential effects of a climate induced drought and increased salinity on aquatic invertebrate communities of a south African lake, and to assess their potential recovery after the end of the drought.
The research is well structured, the experimental plan is presented clearly, and the discussions are coherent and explanatory, but it has a serious flaw concerning the data analysis, in fact, zooplankton cannot be analyzed together with all the other aquatic invertebrates. Zooplankton and aquatic macroinvertebrates represent very different groups of organisms, both by life strategies, by abundance and biomass. These two components can clearly be treated both in the manuscript, but only separately. This aspect regards both the data analysis and for the data presentation (tables and figures).
On the other hand, the planktonic component is represented by a small number of taxa, so it could be easily treated apart.
In addition, it is important to clearly illustrate how much the lake level has dropped during the drought. This is to clearly define how the sampling areas change in the various periods (since the invertebrates were always sampled at a maximum depth of 50 cm).
The manuscript should therefore be revised in the light of these observations.
PARTICULAR COMMENTS
Below I suggest some corrections or additions.
Throughout the paper:
- delete the dot within the unit of measurement, for example replace "g.L-1" with "g L-1".
Abstract
- line 17: specify the correct range of salinity, i.e. “between 9.8 e 11.5 g L-1” not only “> 9 g L-1”.
- Materials and Methods
- specify in the sampling methods if the 40 d-frame dip net sweeps were carried out at each sampling site or overall in the 4 sites at each sampling date, this aspect is not clear
- I would recommend transforming biological data not with the square root, but with the logarithm method which is much more recurrent in this type of data analysis.
- another aspect that is not clear is why the environmental data have been normalized with the logarithmic transformation.
- Results
- in this section, regarding the presentation of data, zooplankton and aquatic macroinvertebrates should be treated separately.
- at the beginning of the results section (lines 247-268) it would be advisable to provide readers with an overview of the communities found over the various years, not only on the families but also on the phyla, and to make references to the overall abundances (not just the diversity) of the nine samplings or 3 years.
- line 250-266: change the names of species and genera in italics.
- lines 301-312: there are no reference tests for these statements (correlation analysis) therefore you cannot write about correlation
- line 306: replace “conductivity” with “salinity”
- Discussion
- as regards the discussion of the data, I suggest once again to divide the two components, as the planktonic one from the benthic one.
- line 383: chironomids do not produce “drought resistance cysts”, amend this statement
- Conclusions
- line 451: “drying” seem to be inappropriate. You should define whether the lake was completely dry or not, see also comment in the general section
Figures
- Figure 2b and 3: add info on the axis (% and coordinates), also if they are already written in the legends.
Supplementary material
- improve the layout of the supplementary material (Tables 1 and 2), now is too confusing.
- Table 1: in the first two columns (species name and family) there are species that are not species and groups that are not families. Replace “species name” with “taxa” and then indicate the family, if possible, and a higher grouping.
Author Response
We thank Reviewer 1 for their valuable input. Please see the attachment with the specific comments made by the reviewer and our responses to them.

Reviewer 2 Report
The manuscript entitled "Aquatic invertebrate community resilience and recovery in response to a supra-seasonal drought in an ecologically important naturally saline lake" is very promising and potentially valuable for many ecologists. The authors have probably taken advantage of sampling the Nyamithi lake prior the supraseasonal drought, continued in sampling and can thus present results of kind of natural experiment. However, it is still an observation survey with all subsequent limitations - I will mention them later.
This research has some limitations in sampling, species identification and results presentation. Nevertheless I think the manuscript is still valuable and brings worth publishing data.
Here are particular issues I see problematic:
1) The sampling of the whole lake is generally insufficient. Only two sites of each biotope (vegetation and substrate) along whole lake is really not much. Also, although I understand reasons why deeper water could not be sampled (hippos and crocodiles are good reason), avoiding sites with depth exceeding 50 cm could result in missing invertebrate taxa.
Of course the sampling cannot be changed, but I suppose the authors could add a paragraph on limitations of the research with mentioning this (and maybe some more) drawback.
2) Another problem is with insufficient determination of the species. Most specimens are identified to the genera or even higher taxa. This is problems in two ways - a) species richness indexes are calculated on the basis of species, or maybe also higher taxa, but same level taxa. Mixing species, genera and families makes interpretation of these indices problematic. Of course e.g. Chironominae larvae in Dec 2014 (n=40) can mean e.g. 8 species, whereas same taxon in Feb 2017 (n=70) can mean 1 dominant species. Computing and discussing species richness without species data is a big problem.
b) many congeneric species differ in salinity tolerance (see e.g. CARBONELL, J.A., MILLÁN, A. and VELASCO, J. (2012), Concordance between realised and fundamental niches in three Iberian Sigara species (Hemiptera: Corixidae) along a gradient of salinity and anionic composition. Freshwater Biology, 57: 2580-2590. https://doi.org/10.1111/fwb.12029). This is again problem in comparing genera and higher level taxa.
It would be great if the authors would be able to determine the specimens into the species level, it would improve the manuscript really much.
3) Authors tried to compare the diversity indices and find significant differences between years 2014, 2016 and 2017. The approach used in this manuscript is wrong due to two reasons:
a) It would be better to use e.g. repeated measures ANOVA instead of multiple t-tests
b) If I understand well, the authors pool all samples from given year (hydrologic period) and compare it using a t-tests. Still, the sample sizes are probably too small (number of indices of particular year (there are missing Degrees of freedom in Table S3)). I suggest omitting tests for significant differences. The charts at fig. 4 are good even without significance tests (actually the way of denoting significant differences is confusing), it is enough to display the differences graphically (number of samples is not sufficient for full statistical analysis).
4) There are some minor drawbacks throughout the manuscript, here I list just some of them, I recommend carefull text correction:
Some species names are not written in Italic (especially p. 6)
Missing word "rapid ???" (l. 421)
Missing "sp." in tab. S1 (Appasus)
The tables in supplement are not formatted properly
Author Response
We appreciate the comments made by Reviewer 2 and thank them for taking the time to review the manuscript. Please see the attachment for the comments made by Reviewer 2 and our response to them.

Reviewer 3 Report
Opinion for authors and Editorial office of "Water"
Title of article: Aquatic invertebrate community resilience and recovery in re-sponse to a supra-seasonal drought in an ecologically important naturally saline lake
Authors: Lizaan de Necker,Johan van Vuren, Luc Brendonck, Victor Wepener and Nico Smit
The article is interesting and worth of publish in "Water" after making some minor changes. I have indicated my suggestions in the attached file. My general comment:
- The introductory chapter should be shortened
- In the discussion, the authors could emphasize which invertebrate formation (zooplankton and benthos) changes during drying out and which regenerates faster.
- Why are some organisms more sensitive and other tolerant on changes as the lake dries up

Author Response
We thank Reviewer 3 for their valuable input into the manuscript. Please see the attachment for the comments made by the reviewer and our responses to them.

Reviewer 4 Report
some comments in the text
- What is the main question addressed by the research? — How does animal diversity in low saline lakes react on extreme climatic events.
- Is it relevant and interesting? — yes
- How original is the topic? — high original
- What does it add to the subject area compared with other published material? — new interesting data on influence of long-term drought on species richness and composition of animals in not previously studied lake.
- Is the paper well written? — yes
- Is the text clear and easy to read? -yes
- Are the conclusions consistent with the evidence and arguments presented? — I think so.
- Do they address the main question posed? -yes.

Author Response
We thank Reviewer 4 for their valuable inputs. Please see the attachment for the specific comments made by the reviewer and our responses to them.

Round 2
Reviewer 1 Report
Manuscript “Aquatic invertebrate community resilience and recovery in response to a supra-seasonal drought in an ecologically important naturally saline lake”
by Lizaan de Necker, Luc Brendonck, Johan van Vuren, Victor Wepener and Nico Smit
The authors of the manuscript answered in a precise manner to all the reviewers’ suggestions; even in the cases where the requests made could not be accepted, the authors' responses were exhaustive.
The text, the figures and the tables in supplementary material were corrected or integrated as required by the reviewers.
The manuscript is now improved in clarity, the investigation and data analysis methods are clearer, and this version has its own consistency.